

# Dietary composition and feeding preference of Mantled guereza *Colobus guereza* (Rüppell, 1835), in Maze National Park, Ethiopia

Abraham Tolcha[1], Matewos Masne[2] and Belayneh Ayechw[2]

[1] Biodiversity Research and Conservation Center, College of Natural and Computational Sciences, Arba Minch University, Arba Minch, Southern, Ethiopia
[2] Department of Biology, College of Natural and Computational Sciences, Arba Minch University, Arba Minch, Southern, Ethiopia

## ABSTRACT

Knowledge of feeding ecology is essential for effective management of a primate and its habitat. The Mantled guereza *Colobus guereza* is a predominantly folivorous monkey that occurs in different parts of eastern Africa, including the Maze National Park in Ethiopia. Despite many studies conducted in the area, there is no up-to-date data that was carried out on feeding ecology of the *Colobus guereza*. The aim of this study is to determine the dietary composition and feeding preference of the *Colobus guereza* in the park. To better understand this, we randomly selected three study groups along the Maze River. We used instantaneous scan sampling method to collect feeding data from September 2021–August 2022. We followed guerezas from 6:30 to 10:30 in the morning and 13:30 to 17:25 in the afternoon collecting feeding activity data between 5 min intervals during 10-min scan duration. Overall, guerezas were observed to eat eight plant species and unidentified invertebrates in the park. Of these, *Trichilia emetica* contributed the highest proportion accounted 53.36% and 27.83% in the wet and dry season respectively, while unidentified invertebrates were rarely utilized over the course of this study. We also found that young leaves were consumed more ($n = 1{,}794$, 75.31%) in the wet while mature leaves were eaten more ($n = 1{,}215$, 43.61%) over the other diet components in the dry season. These results suggest that the *guerezas* in the park exhibit temporal dietary flexibility. The observed dietary flexibility may be partly due to seasonal changes in availability of food plant parts in the groups' home ranges in the park. Our results suggest that maintaining the park is critical to protect food plant species for this primate, which at present constitutes only a few.

## INTRODUCTION

Habitat change and climate change are significantly limiting species' access to essential food sources (*Terborgh, 2015*; *Ellsworth, 2017*), and leading to biodiversity loss (*Maestre et al., 2012*). Understanding species' dietary composition and preferences is fundamental

Corresponding author
Abraham Tolcha,
abrahamtolcha2@gmail.com

for guiding the development of sound conservation practices for a species and its habitat (*Ramesh & Downs, 2013*). Feeding ecology studies can be used to identify crucial food resources and their spatio-temporal availability (*Sengupta, McConkey & Radhakrishna, 2015*) and to quantify the effects of habitat change, loss and fragmentation on animal populations (*Irwin et al., 2014*).

The ongoing habitat modification due to a variety of anthropogenic pressures (*Estrada et al., 2017*, *2020*; *Estrada & Garber, 2022*; *Garber, 2022*), and climate change provides a strong premise for studying diet composition and food preference in primate species. For instance, about 65% of primate species are threatened with extinction, and ~75% have declining populations as a result of persistent human pressures on natural environments leading to widespread loss and degradation of tropical forests (*Estrada et al., 2017*; *Mittermeier et al., 2022*). Habitat loss and degradation result in loss or decline of important food plant species for primates (*Estrada et al., 2017*; *de Paula Mateus et al., 2018*; *Mekonnen et al., 2020*), and this may eventually drive a primate species into extinction. Even those primate species that occupy protected areas like national parks are equally affected by climate change. Climate change could affect availability of primates' food resources in part by altering phenological patterns of some food plant species (*Pinto et al., 2023*). The effect of climate change provides a strong basis for studying feeding ecology for primate species in protected areas in order to provide baseline feeding data that can be monitored in the future.

Primates feed on a diverse array of plant items and animal tissues to meet their nutritional needs (*Coiner-Collier et al., 2016*). In response to habitat changes, they can develop ecological and behavioural flexibility (*Arroyo-Rodríguez & Fahrig, 2014*; *Mekonnen et al., 2018*). Studies indicate that species exhibit microhabitat preferences, occupying specific forest strata or habitat types (*Campbell et al., 2019*; *Matsuda et al., 2022*) to exploit various resources that meet their nutritional requirements. The spatial and temporal resource availability is among the factors which can determine the distribution of a primate species (*Mendoza-Soto et al., 2024*). Food availability in an animal's diet is influenced by seasonal variations among other environmental factors (*Chouteau, 2006*). Some primate food resources, for instance young leaves, decline in dry season and this may compel folivorous primates to include more barks and mature leaves in their diet (*Arseneau-Robar et al., 2021*). Dietary shifts typically correspond with seasonal resource scarcity (*Yiming, 2006*; *Hanya & Chapman, 2013*) and probably seasonal changes in chemical composition of food plant species (*Matsuda et al., 2022*; *Ravhuhali, Msiza & Mudau, 2022*). Thus, a shift in an individual's diet should reflect the most profitable foods available at a specific time and place, which may also mean the most nutritious, the easiest to find, or the easiest to process (*Lambert & Rothman, 2015*).

Primates are among the most endangered mammals (*Sushma, Ramesh & Kumara, 2022*), facing significant threats from habitat loss (*Kifle & Beehner, 2022*), hunting, and the illegal pet trade (*Cowlishaw & Dunbar, 2004*; *Ripple et al., 2016*; *Estrada et al., 2017*). While their general feeding ecology is well understood, it is important to recognize that feeding ecology is highly site-specific and species-specific, influenced by factors such as local vegetation, seasonal resource availability, and competition with other species

(*Chapman & Balko, 2002*; *Estrada et al., 2017*). To guide ecological restoration efforts and inform sustainable forest management practices, we need site-specific information to ensure the availability of critical food resources for primates (*Ganzhorn et al., 2017*). The *Colobus guereza* is a Least Concern species by IUCN (*de Jong, Butynski & Oates, 2019*), and occurs in different parts of equatorial Africa, including Ethiopia. In Ethiopia, the species was reported to be present in the Maze National Park by *Dansa & Tekalign (2022)*. It feeds mainly on leaves (*Harris & Chapman, 2007*; *Matsuda et al., 2020*). The amount of different plant parts eaten vary among groups and seasons (*Harris & Chapman 2007*; *Hussein, Afework & Dereje, 2017*; *Matsuda et al., 2020*). Despite many studies on its feeding ecology on different parts of its geographical range, the species was not studied in the Maze National Park up to present.

The aim of this study was to determine dietary composition, and feeding preferences of *C. guereza* in the park. Here, we hypothesized that seasonal change affects food availability, which in turn determines the dietary composition and feeding preference of the study species. Our findings suggest that season affects the accessibility of diet components and consequently influence feeding preferences of the *C. guereza*. This study is expected to offer an opportunity to create and implement successful habitat conservation strategies to preserve important food resources in the Park.

## METHODS AND MATERIALS

### Study area

We conducted this study at Maze National Park (MzNP) along the Maze River, a critical habitat for our target species. This park is located in a bio-diverse region of southern Ethiopia, situated between the Gamo and Gofa Zones. It provides a unique and varied landscape that supports a rich array of flora and fauna. The park is situated between 6°18′30″ to 6°29′00″ N latitude and 37°7′30″ to 37°22′30″ E longitude, positioning it within a specific ecological and climatic zone of Ethiopia. The elevation ranges from 900 to 1,200 m above sea level, which influences the climate, vegetation, and biodiversity (*Befekadu & Afework, 2006*). The annual rainfall varies between 843 mm and 1,321 mm, which reflect variability in precipitation. The Maze area's climate exhibits a distinct seasonal pattern characterized by a rainy season from March to October and a dry season from November to February. The lowest temperature during the rainy season is 15.3 °C in June, suggesting relatively mild temperatures, which may be beneficial for plant growth and biodiversity during this time, and the highest temperature during the dry season is 33.5 °C in February (*Mamo, 2012*), indicating hotter conditions that could lead to water stress for plants and increased evaporation rates.

The park is home to a remarkable variety of mammalian fauna (*e.g.*, orbi *Ourebi aourebi*, bohor red buck *Redunca redunca*, buffalo *Syncerus caffer*, warthog *Phacochoerusafricanus*, bush buck *Tragelaphus scriptus*, greater kudu (*Tragelaphus strepsiceros*), lesser kudu (*Tragelaphus imberbis*), Water buck *Kobus ellipsiprymnus*, bush pig *Potamocherus larvatus*, anubus baboon *Papio anubis*, vervet monkey *Chlorocebus pygerythrus*, Mantled guereza *Colobus guereza*, lion *Panthera leo* leopard *Panthera pardus*,

wildcat *Felis silvestris*, and serval cat *Leptailurus serval*, in addition to varied floral composition. It also comprises varieties of bird species, reptiles, amphibians and insects.

Maze National Park (MzNP) is predominantly covered by savannah grassland interspersed with scattered deciduous broad-leaved trees, creating a diverse landscape that supports a variety of wildlife. The majority of the park consists of plains enriched by open vegetation dominated by Combretum and Terminalia species, which play a critical role in providing habitat and food sources for the local fauna. The Maze River, originating from the surrounding highlands, flows along with its numerous tributaries, traverses the park from the northern to the southern end, creating vital riparian zones that serve as corridors for wildlife movement and habitat for aquatic and semi-aquatic species. This unique combination of savannah grasslands and riverine habitats makes MzNP an important area for our target species.

The Maze River is essential to the park's biological well-being because it forms verdant riparian areas that provide a variety of species with places to reproduce and feed. Given an important habitat to primates, particularly for guerezas, in which no feeding activity has been detected rather than the riverine habitat over the study period (Fig. 1).

## Study groups

Three groups of *C. guereza* were targeted for this study. One group with three individuals at Maze camp site (group-1); another group with two individuals at Domba site (group-2); and the third group with three individuals at Lemasse site (group-3); were randomly selected along the River Maze. We monitored those groups in their home ranges for the duration of the study, with a research team assigned to each group to look at dietary ecology and potential differences in feeding activities.

## Data collection

### Plant phenology and dietary preferences

During reconnaissance before starting actual data collection for this study, we made marking for individual plants (trees, shrubs) with DBH ≥ 10 cm. To assess food availability, we collected vegetation data within the three groups' home range using randomly located nine plots of 20 m × 20 m for large trees *i.e.*, ≥10 cm in diameter at breast height, DBH). We placed the plots within the home range of each study group and we carefully observed all plant parts being used as food sources for each month over the study period. Each month, all stems in every plot were carefully examined through the study year and assigned to one of its food items (young leaf, mature leaf, fruit, bark, shoot and flower). We computed selection ratios for each plant species that was observed to have been fed by the *Colobus guereza*. Values close to zero indicate rejection, implies the plant species was eaten less than would be expected based on its availability, while large values show preference selected more than predicted based on availability. Thus, the avoidance of an item does not to indicate the item is never consumed and does not mean a non-food item (*Clink et al., 2017*). We sampled a total of nine plots, each accounted 20 m × 20 m = 400 m², *i.e.*, a total area of (400 m² × 9 = 3,600 m²), three plots for each site (Maze camp site, Domba site, Lemasse site) within the home ranges' of three randomly selected study groups along the Maze River. Due

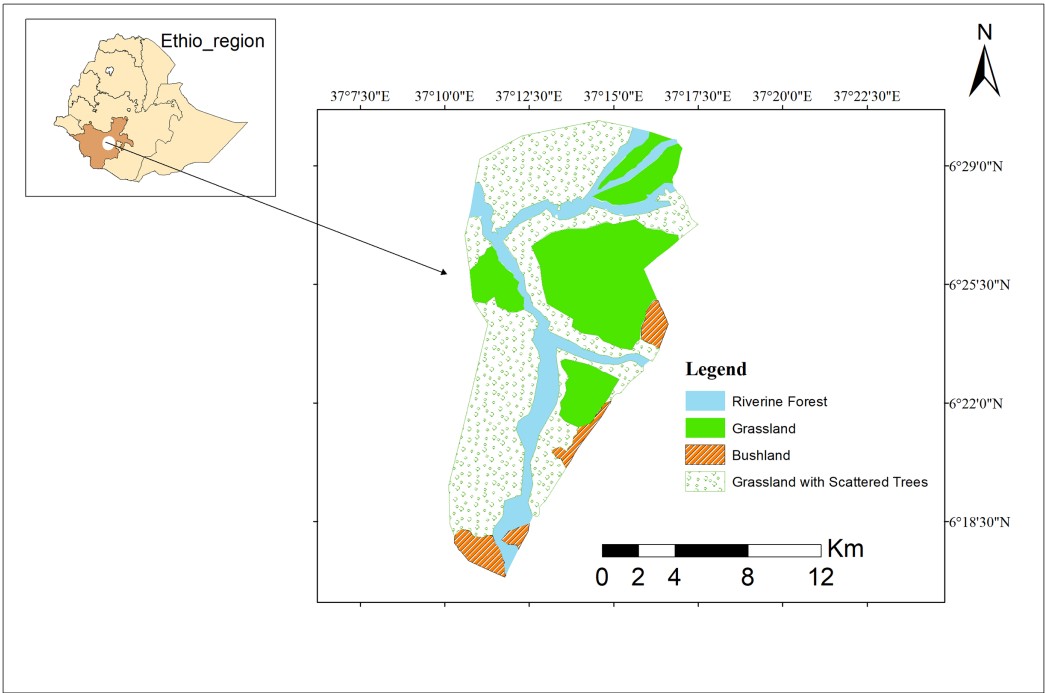

**Figure 1 Study area map.** Study area map of Maze National Park. Image credit: ArcGIS.

to the homogeneity of the habitat, there is no need to compare the dietary composition differences between the groups. Most plant species were identified in the field, while those that could not be identified were collected and later brought to Arba Minch University for further examination and taxonomic identification by experts. For the analysis of food availability, we used 12 months of phenological data concerning the eight plant species that were most frequently consumed by the study species.

### Diet composition

We collected feeding data for 12-month period between September 2021 to August 2022 using instantaneous scan sampling (*Altmann, 1974*, *2009*). For the dry season, we collected data from September 2021 to February 2022, and for the wet season, from March 2022 to August 2022. The feeding data were collected through direct observation using binoculars from designated viewpoints (*Altmann, 2009*). Observations were conducted for a fixed period of 10 min, with 5-min intervals, during the morning from 6:30 to 10:30 and in the afternoon from 13:30 to 17:25 (*Fashing, 2007*). During scans, the plant species, plant parts, growth forms, and other animals consumed were recorded (*Fashing et al., 2014*; *Jarvey et al., 2018*; *Mekonnen et al., 2018*). We categorized the food components as Young leaf: a newly grown leaf that is still developing and has not yet reached its full maturity, often smaller, softer, and lighter in color compared to mature leaf; Mature leaf: a fully developed leaf that has reached its maximum size and structural maturity, typically tougher, darker in color compared to young leaf; Fruit: the reproductive structure of a plant that contains the seeds, ripe and often eaten by animals; Bark: the protective outer layer of the stem or trunk

of a woody plant, and composed of multiple layers, not to mean dead layers but, living layers; Shoot: the aboveground vegetative part of a plant includes the stem and buds; Flower: the reproductive structure of a plant, and are responsible for sexual reproduction in plants; *Unidentified invertebrates*: Small, non-vertebrate animals that could not be identified to a specific taxonomic level, particularly insects. We then compared the number of feeding observations for each food items.

## Data analysis

We combined feeding data from the three groups into one dataset before the computations of proportions of each diet component. The analyses were executed in XLSTAT 2023.1.3 (1,407) and SPSS software version 22. Of the total of 11,520 scans, we recorded 5,168 (44.86%) feeding activities over the study period, (Wet: 2,382, 46.09%; Dry: 2,786, 53.91%). We computed the proportion of the diet components for seven food classes (young leaf, mature leaf, fruit, bark, shoot, flower, unidentified invertebrates) recorded through the study period by dividing the number of records of a particular diet component by the total number records from all diet components. The proportion of each diet component was then converted into percentages. The chi-square test was employed to test for the seasonal and monthly variations in proportions of the diet components. To assess the feeding preferences of various food plant species in a given habitat, we computed the selection ratio for each species. Stem density measurement for each food plant species present in the study area, we calculated the total stem density, which represents the number of stems per unit area. We also determined the percentage of each food plant species relative to the total number of food plants in the study area.

**Computation of selection ratio:** The selection ratio (SR) for each species was calculated using the formula: SR = Percentage of Species/Stem Density Percentage.

Where: The percentage of a certain food plant species among all food plants is referred to as the percentage of species. The stem density percentage describes the ratio of the stem density of a particular food plant species to the total stem density of all species.

Thus, a species is preferred over others in the habitat if the selection ratio is greater than 1. A species may be avoided if its selection ratio is smaller than 1, which indicates that it is less desirable. A selection ratio equal to 1 indicates neutral selection, meaning the species is utilized in proportion to its availability.

## Field permit

The Office of Executive Research Directorate and the Biodiversity Research and Conservation Center, Arba Minch University were approved the fieldwork under research permit (AMU/TH2/BRCC/09/2014). Hereby, we can guarantee that no animal capture and tissue or blood sample was taken from the subject species, as data were recorded through direct observation without animal capture.

# RESULTS

## Food plant species consumption and preferences

We found that six tree plants (*Acacia polyacantha*, *Millettia ferruginea*, *Moringa stenopetala*, *Syzygium guineense*, *Trichilia emetica* and *Ficus sycomorus*) and two shrub
**Table 1 Feeding preference.** Feeding preference (selection ratio) of food plant species consumed by *Colobus guereza* during the study period.

| Family | Species | % of food plant species | % stem density Stem/ha | Selection ratio | Rank |
|---|---|---|---|---|---|
| Apocynaceae | *Carissa spinarum* Shrub | 4.22 | 7.22 | 0.58 | 6 |
| Fabaceae | *Millettia ferruginea* Tree | 2.32 | 4.16 | 0.56 | 7 |
| Myrtaceae | *Syzygium guineense* Tree | 17.92 | 5 | 3.58 | 2 |
| Moringaceae | *Moringa stenopetala* Tree | 7.64 | 4.16 | 1.84 | 4 |
| Meliaceae | *Trichilia emetica* Tree | 39.59 | 9.16 | 4.32 | 1 |
| Moraceae | *Ficus sycomorus* Tree | 0.81 | 4.16 | 0.19 | 8 |
| Malvaceae | *Grewia villosa* Shrub | 21.01 | 9.44 | 2.22 | 3 |
| Fabaceae | *Acacia polyacantha* Tree | 5.24 | 4.44 | 1.18 | 5 |

**Table 2 Diet components.** The proportion of dietary components from different plant species and non-plant sources consumed during the wet season (March to August) and the dry season (September to February) throughout the study period.

| Food items used | Food components consumed during wet and dry seasons (%) | | | | | | | | | | | | | |
|---|---|---|---|---|---|---|---|---|---|---|---|---|---|---|
| | Wet | | | | | | Total | Dry | | | | | | Total |
| | YL | ML | FR | Bk | Sh | FL | | YL | ML | FR | Bk | Sh | FL | |
| *Carissa spinarum* | 0 | 0 | 6.04 | 0 | 0 | 0 | 6.04 | 0 | 0 | 2.67 | 0 | 0 | 0 | 2.67 |
| *Millettia ferruginea* | 0 | 0 | 0 | 0 | 0 | 0 | 0 | 0.06 | 0 | 1.83 | 2.40 | 0 | 0.06 | 4.35 |
| *Syzygium guineense* | 12.93 | 0 | 0 | 0 | 0 | 0 | 12.93 | 1.90 | 10.37 | 3.30 | 0 | 0.1 | 6.49 | 22.16 |
| *Moringa stenopetala* | 0.38 | 0 | 0 | 0 | 0 | 0 | 0.38 | 13.86 | 0 | 0 | 0 | 0 | 0 | 13.86 |
| *Trichilia emetica* | 39.80 | 5.37 | 0 | 0 | 8.19 | 0 | 53.36 | 6.44 | 17.30 | 3.55 | 0 | 0.54 | 0 | 27.83 |
| *Ficus sycomorus* | 0 | 0 | 0 | 0 | 0 | 0 | 0 | 0 | 0 | 1.5 | 0 | 0 | 0 | 1.5 |
| *Grewia villosa* | 15.57 | 4.75 | 0 | 0 | 0 | 0 | 20.32 | 5.57 | 15.94 | 0 | 0 | 0 | 0.1 | 21.61 |
| *Acacia polyacantha* | 6.63 | 0 | 0 | 0 | 0 | 0 | 6.63 | 0 | 0 | 0 | 4.05 | 0 | 0 | 4.05 |
| Unidentified invertebrates | – | – | – | – | – | – | 0.34 | – | – | – | – | – | – | 1.97 |
| Total | 75.31 | 10.12 | 6.04 | 0 | 8.19 | 0 | 100 | 27.82 | 43.61 | 12.85 | 6.45 | 0.64 | 6.65 | 100 |

species (*Carissa spinarum*, *Grewia villosa*) that were grouped under seven families (Apocynaceae, Fabaceae, Myrtaceae, Moringaceae, Meliaceae, Moraceae and Malvaceae) were consumed by *C. guereza* in the study site. Except for the Fabaceae, all are represented by a single species. Overall, *Trichilia emitica* was the most top plant species preferred to the rest (Table 1).

During the wet season, *Trichilia emetica* contributed the largest proportion to the total amount of young leaf consumption, accounting for 52.85% ($n = 948$). This was followed by *Grewia villosa* at 20.68% ($n = 371$), while *Moringa stenopetala* was rarely reported, with only 0.5% contributions ($n = 9$) (Table 2). The second most popular food item this season was mature leaves, with *Trichilia emetica* and *Grewia villosa* making up the largest portions making up 53.12% and 46.88% respectively (Table 2). During the dry season, the *Moringa stenopetala* contributed the largest portion (386, 49.87%) of all young leaf consumption and *Millettia ferruginea* contributed the least (0.13%) (Table 2).

**Table 3 Percentage composition.** Percentage composition of annual and seasonal dietary composition.

| Plant parts eaten | YL | ML | FR | Bk | Sh | Fl | Unidentified inveretebrates |
|---|---|---|---|---|---|---|---|
| Wet season | 75.31 | 10.12 | 6.06 | 0 | 8.19 | 0 | 0.34 |
| Dry season | 27.82 | 43.82 | 12.85 | 6.45 | 0.64 | 6.65 | 1.97 |
| Annual/Overall | 51.57 ± 23.7 | 26.97 ± 16.9 | 9.46 ± 3.4 | 3.22 ± 3.2 | 4.42 ± 3.7 | 3.32 ± 3.3 | 1.16 ± 0.8 |

**Table 4 Proportion of diet components.** Proportion of diet components used by *Colobus guerezas* for each month during the study period.

| No. | Food components | Diet components consumed by *Colobus guerezas* over months of the year (%) | | | | | | | | | | | |
|---|---|---|---|---|---|---|---|---|---|---|---|---|---|
| | | Nov. 2021 | Dec. 2021 | Jan. 2022 | Feb. 2022 | Mar. 2022 | Apr. 2022 | May 2022 | June 2022 | July 2022 | Aug. 2022 | Sep. 2022 | Oct. 2022 |
| 1 | Young leaf | 31.64 | 25.27 | 30.45 | 29.83 | 75.12 | 76.88 | 70.48 | 76.07 | 77.14 | 76.6 | 20.55 | 28.66 |
| 2 | Matured leaf | 37.39 | 41.05 | 39.53 | 39.91 | 9.67 | 6.99 | 11.19 | 9.07 | 8.57 | 15.6 | 53.2 | 50.61 |
| 3 | Fruit | 17.92 | 18.53 | 11.88 | 15.45 | 6.52 | 8.06 | 8.57 | 6.05 | 6.43 | 0 | 11.19 | 6.91 |
| 4 | Bark | 4.2 | 5.46 | 6.48 | 5.58 | 0 | 0 | 0 | 0 | 0 | 0 | 5.7 | 4.87 |
| 5 | Shoot | 0.45 | 3.37 | 1.08 | 0.43 | 7.49 | 8.06 | 9.05 | 8.81 | 7.86 | 7.8 | 0 | 0.61 |
| 6 | Flower | 6.64 | 4.42 | 8.64 | 5.37 | 0 | 0 | 0 | 0 | 0 | 0 | 7.54 | 7.32 |
| 7 | *Unidentified invertebrates* | 1.77 | 1.9 | 1.94 | 3.43 | 1.2 | 0 | 0.74 | 0 | 0 | 0 | 1.82 | 1.02 |

## Diet composition of guerezas

The annual diet of *C. guereza* comprised of young leaf, matured leaf, fruit, bark, shoot and flower (Table 3). Young leaf was the most consumed plant part in the overall annual diet. Based on seasons, there was seasonal variation in number of feeding records of all food plant items (Young leaf: $\chi^2 = 405.140$, df = 1, $p < 0.05$; Mature leaf: $\chi^2 = 651.563$, df = 1, $p < 0.05$; Fruit: $\chi^2 = 105.593$, df = 1, $p < 0.05$; Shoot: $\chi^2 = 125.063$, df = 1, $p < 0.05$). Young leaf as the major food item was more frequently consumed in the wet season ($n = 1,794$, 75.31%) while mature leaves were consumed more over the other food components during the dry ($n = 1,215$, 43.61%) than wet season (Table 3). Interestingly, consumption of unidentified invertebrates was also recorded to increase by about 1.63% in the dry season (Table 3). The results also demonstrate some monthly variations in consumption of different plant parts or diet components by guerezas (Table 4).

## DISCUSSION

### Food plant species consumption and preferences

Comparable to other studies on feeding ecology of *Colobus guereza* across its geographical range, our study reports very few plant species consumed by this primate. In this study, we recorded eight plant species in the Maze National Park which are fewer than that observed in Kalinzu Forest (39 plant species) by *Matsuda et al. (2020)*. Of these eight species, only two plant species *Trichilia emitica* (39.6%) and *Grewia villosa* (21.01%) had the highest feeding records and thus dominate monkeys' diet. The observation of few plant species eaten by guerezas suggests that the dietary plant richness of this primate is very low in the

**Table 5 Plant species composition.** Plant species composition along trials of study species' home range.

| Family | Species | Growth form | Maze camp site (Group-1) | | | Domba site (Group-2) | | | Lemasse site (Group-3) | | | Total |
|---|---|---|---|---|---|---|---|---|---|---|---|---|
| | | | $P_1$ | $P_2$ | $P_3$ | $P_4$ | $P_5$ | $P_6$ | $P_7$ | $P_8$ | $P_9$ | |
| Apocynaceae | *Carissa spinarum* | Shrub | 4 | 2 | 0 | 5 | 1 | 4 | 2 | 0 | 8 | 26 |
| Fabaceae | *Millettia ferruginea* | Tree | 3 | 0 | 4 | 0 | 3 | 0 | 4 | 0 | 1 | 15 |
| Fabaceae | *Tamarindus indica* | Tree | 2 | 5 | 0 | 0 | 1 | 0 | 3 | 2 | 0 | 13 |
| Balanitaceae | *Balanites aegyptiaca* | Tree | 0 | 1 | 3 | 4 | 1 | 0 | 1 | 1 | 0 | 11 |
| Anacardiaceae | *Rhus glutinosa* | Tree | 1 | 0 | 0 | 1 | 0 | 2 | 0 | 2 | 0 | 6 |
| Ebenaceae | *Diospyros abyssinica* | Tree | 1 | 0 | 0 | 2 | 0 | 1 | 0 | 1 | 0 | 5 |
| Salicaceae | *Flacourtia indica* | Tree | 0 | 0 | 2 | 0 | 4 | 0 | 1 | 0 | 0 | 7 |
| Olacaceae | *Ximenia americana* | Tree | 0 | 2 | 1 | 0 | 0 | 1 | 0 | 0 | 2 | 6 |
| Myrtaceae | *Syzygium guineense* | Tree | 4 | 2 | 0 | 3 | 1 | 0 | 2 | 5 | 1 | 18 |
| Moringaceae | *Moringa stenopetala* | Tree | 3 | 2 | 1 | 2 | 1 | 0 | 2 | 0 | 4 | 15 |
| Fabaceae | *Piliostigma thonningii* | Tree | 0 | 0 | 3 | 1 | 0 | 0 | 5 | 2 | 0 | 11 |
| Meliaceae | *Trichilia emetica* | Tree | 2 | 5 | 1 | 5 | 3 | 8 | 4 | 3 | 7 | 38 |
| Moraceae | *Ficus sycomorus* | Tree | 1 | 4 | 0 | 1 | 2 | 3 | 1 | 2 | 1 | 15 |
| Rhamnaceae | *Ziziphus spina-christi* | Tree | 0 | 3 | 2 | 0 | 0 | 2 | 3 | 1 | 2 | 13 |
| Malvaceae | *Grewia villosa* | Shrub | 3 | 6 | 1 | 2 | 5 | 8 | 7 | 0 | 2 | 34 |
| Rubiaceae | *Gardenia ternifolia* | Tree | 0 | 1 | 1 | 2 | 0 | 0 | 1 | 2 | 0 | 7 |
| Anacardiaceae | *Sclerocarya birrea* | Tree | 2 | 1 | 0 | 2 | 0 | 1 | 3 | 0 | 0 | 9 |
| Combretaceae | *Terminalia brownii* | Tree | 1 | 0 | 4 | 0 | 1 | 3 | 0 | 1 | 2 | 12 |
| Malvaceae | *Grewia mollis* | Tree | 2 | 1 | 0 | 2 | 0 | 2 | 1 | 0 | 2 | 10 |
| Rutaceae | *Harrisonia abyssinica* | Tree | 0 | 0 | 2 | 1 | 0 | 1 | 1 | 0 | 1 | 6 |
| Fabaceae | *Acacia polyacantha* | Tree | 1 | 3 | 0 | 0 | 2 | 4 | 1 | 2 | 3 | 16 |
| Sub-total | | | 30 | 38 | 25 | 33 | 25 | 40 | 42 | 24 | 36 | 293 |
| Total | | | 93 | | | 98 | | | 102 | | | |

park. This provides an urgent need to conserve the park to ensure the long-term presence of important food plant species. It appears that the guerezas in MzNP consumes food plant species as expected from its availability across its home range. Most of the plant species preferred (having high selection ratios) are those which are quite abundant in the groups's home ranges (Tables 1 and 5). However, this does not necessarily indicate that they are the most nutritious or preferred food sources, but rather they are fed because they are quite abundant in the habitat and not because they are most nutritious. Future studies should analyze nutrient content and other phytochemical composition of plants eaten in order to draw decisive conclusion on plant food preferences.

Our study demonstrates a seasonal variation in frequency with which certain plant species were eaten. For example, *Trichilia emitica* was most frequently eaten during the dry season while *Syzygium guinense* and *Grewia villosa* were eaten in the dry season. Seasonal variation in plant phenological patterns can in part explain the observed variation in feeding plant species between seasons. We observed that monthly dietary diversity

increased as the number of available plants with young leaves reduced during the dry season (*Kibaja et al., 2023*) (Table 2). For example, the plant species such as *Ficus sycomorus*, *Millettia ferruginea* and *Moringa stenopetala* were not used as food source for the study species during the wet season, because of sufficient young leaves, with exceptions thus *Moringa stenopetala* only contributed small amount in May (Table S2). On the other hand, *Moringa stenopetala* significantly contributed to the study species bearing more young leaves during the dry season (Table 2). This way, eight plant species from seven families and one non-plant source, *i.e.*, unidentified invertebrates offered food items to the *C. guereza* in the study area. This was particularly due to the effects of the declining availability of young leaves from *Trichilia emetica* and *Grewia villosa*. Much of the dietary diversity in the study group is seemingly attributable to the availability of young leaf portion of their diet.

Many colobine species, have increased dietary extent during times and areas with low availability or quality of resources (*Hu, 2011*; *Clink et al., 2017*). The present study depicted, the dietary extent increased with decreasing in young leaf availability during the dry season.

### Food plant parts/item consumption

We found that the guerezas exploited different plant parts, leaves being mostly eaten in the MzNP. However, it is not surprising for them to consume mostly young leaves because these monkeys like other colobines are anatomically adapted to feed on leaves that facilitate their leaf-eating habits, including their preference for young leaves due to their nutritional benefits (*e.g.*, *Gonzalez & McGraw, 2021*; *Mola et al., 2022*). In line with this, studies show that leaves accounted for high proportion (42–49%) by folivorous-frugivorous monkeys (*Lima & Bicca-Marques, 2024*). Another study has shown that Bale monkey, a folivore specialist, spend more time munching on new bamboo tree leaves in Southern Ethiopia (*Mekonnen et al., 2018*). Similarly, the leaves accounted for highest proportion of *Colobus guereza's* food items (71.6%) in Borena-Sayint National Park, Northern Ethiopia (*Hussein, Afework & Dereje, 2017*) and 82% in Bale Mountains National Park, Ethiopia (*Petros et al., 2018*).

Furthermore, young leaves were highly eaten compared to mature leaves. This observation is in line with a study on feeding ecology of guerezas at Saja Forest, Kaffa Zone, Southwest Ethiopia, that reported the monkeys to eat young leaves over mature leaves (*Mola et al., 2022*). Similarly, *Matsuda et al. (2020)* reported the *C. guereza* to consume up to 87% young leaves in the Kalinzu Forest in Uganda. Young leaves are preferred because they have low fiber content, high nutrients and are easier to digest (*Leighton, 1993*; *Matsuda et al., 2020*). Thus, by preferentially consuming these food items, the guerezas are able to maximize their nutrient intake while minimizing the ingestion of toxic compounds. Interestingly, the guerezas were observed to increase the consumption of invertebrates during the dry season by 1.63% (Table 2). The high consumption of invertebrates during the dry season could be strategy to increase intake of proteins from invertebrates rather than getting it from young leaves which were slowly eaten in this season.

The results of this study have demonstrated some seasonal dietary flexibility for the guerezas in the study site. We observed the study species use young leaves and matured leaves interchangeably during the wet and dry seasons. They consume a lot of young leaves during the wet season and mature leaves during the dry season and vice versa. Throughout the study months, there were considerable changes in availability and consumption rate of diet items (Table 5). This is attributed to seasonal variations in phenological patterns that affect the availability of food items which eventually influence seasonal dietary composition for the guerezas. For instance, in the field, we observed that when young leaves were insufficient during the dry season, hence the monkeys change their diet use by increasing consumption of mature leaves. This is consistent with the previous study where resource availability is highly variable; folivorous monkeys eat more leaves during periods of low fruit availability (*Hanya & Bernard, 2012*). Research findings found, proboscis monkeys varied in response to monthly changes in food availability, but did not vary among forest types (*Feilen & Marshall, 2020*). In addition, the influence of seasonality on the diet reported at Tanjung Putting National Park, thus fruits comprised high proportion of the diet from January to May, while young leaves consumed the highest proportion of the diets from June to December (*Yeager, 1989*). This might be attributed the fact, that the season contribute to the availability and even the quality of diet components and this drives the flexibility for feeding of the species. However, *Colobus guereza* consumed high amount of young leaf during the study period, in riverine habitat of the park.

The results of this study demonstrate that the guerezas exhibit seasonal and monthly dietary variability in response to availability of food components across months or seasons (Tables 4 and 5). Dietary flexibility is a strategy that enables primates to survive during periods of food shortage (*Feilen & Marshall, 2020*) or exploit different parts having different food resources across their home ranges or habitats (*Sha & Hanya, 2013*; *Seiler & Robbins, 2016*; *Mekonnen et al., 2018*; *Tesfaye et al., 2021*).

## CONCLUSION

The results of this study demonstrate low richness of dietary plant species for guerezas in the park. The observation of only eight plant species with only two mostly eaten by the monkeys provide impetus for effective protection of the park to ensure the long-term presence of important food plant species. The reliance of this primate on few plant species gives a daunting future to the survival of this population in the face of ongoing climate change. However, seasonal dietary flexibility in plant species and food plant items (plant parts and invertebrates) provide some promising future as this observation suggest that the primate can respond to habitat changes through ecological flexibility. Our research showed that the habitat found in rivers plays a significant role containing all essential food plants and making a suitable place for the species to reside. We found that plant species' parts, particularly leaves *i.e.*, young and mature, are a fundamental diet items to the *Colobus guereza*. Plant species such as *Trichilia emetica*, *Grewia villosa*, *Syzygium guineense* and *Moringa stenopetala* were reported to be among the most important food sources for providing sufficient leaves (young, mature) to the subject species over the study period,

and we suggest that they be conserved for sustainable conservation of the species. Overall, we strongly recommend that the protection of the riverine habitat will result in effective conservation of *Colobus guereza* and its habitat in the Maze National Park.

## ACKNOWLEDGEMENTS

Our thanks go to all staff members of Maze National Park, for their cooperation and support throughout this work.

### Funding

This research is funded by the Biodiversity research and conservation center, Arba Minch University, Ethiopia (No. GOV/AMU/TH2/BRCC/09/2014). The funders had no role in study design, data collection and analysis, decision to publish, or preparation of the manuscript.

### Grant Disclosures

The following grant information was disclosed by the authors:
Biodiversity Research and Conservation Center, Arba Minch University, Ethiopia: GOV/AMU/TH2/BRCC/09/2014.

### Competing Interests

The authors declare that they have no competing interests.

### Author Contributions

- Abraham Tolcha conceived and designed the experiments, performed the experiments, analyzed the data, prepared figures and/or tables, authored or reviewed drafts of the article, and approved the final draft.
- Matewos Masne performed the experiments, analyzed the data, prepared figures and/or tables, and approved the final draft.
- Belayneh Ayechw performed the experiments, analyzed the data, prepared figures and/or tables, and approved the final draft.

### Field Study Permissions

The following information was supplied relating to field study approvals (*i.e.*, approving body and any reference numbers):

The Office of Executive Research Directorate and the Biodiversity Research and Conservation Center, Arba Minch University approved the study (AMU/TH2/BRCC/09/2014).

### Data Availability

Raw data is available in the Supplemental Files.

## Supplemental Information

Supplemental information for this article can be found online at http://dx.doi.org/10.7717/peerj.18998#supplemental-information.

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
