# Peer review of "Dietary composition and feeding preference of Mantled guereza Colobus guereza (Rüppell, 1835), in Maze National Park, Ethiopia"

_PeerJ, doi:10.7717/peerj.18998_

## Round 0.1 · original submission · Major Revisions

The manuscript shares useful data on the flexibility of the feeding behaviour of Colobus guereza in Maze National Park, Southern Ethiopia. However, before the manuscript can be considered for publication, further details are required to clarify how the data were collected, and re-analysis is needed to interpret the findings properly. Reviewer 1 discusses statistical options but I agree with Reviewer 2 that analyses should be used that can account for interdependencies within the data. Depending on the data analysed, the authors might wish to consider a mixed model approach. Both reviewers refer to external support to improve the paper, but I'm aware this may not be available. As a western researcher, I firmly believe we need to do more to support a more global community of primatologists, especially in-country researchers, and so I would be happy to answer specific questions over email ([email protected]) if further clarification is needed. I'm mindful that publishing in a second language is challenging and translation services can be a costly barrier to publishing. If useful, translatesciences.com have created a list of resources to try to help overcome language barriers in science. Both reviewers have taken time to provide detailed feedback in their reviews which will improve the manuscript and strength of the research.

·

Basic reporting

The English in the entire manuscript should be improved to ensure that an international audience can clearly understand your text (from the abstract to the conclusion). I suggest a fluent English speaker can do this as I am not an English Speaker. Coherence should be maintained throughout the manuscript from the Abstract to Conclusion. Though some references have been mentioned, authors can include some important references specific for guerezas.

Experimental design

The overall aim is stated. However specific aims should be clearly stated, research gap is also not well stated in the introduction. Methods should be comprehensively described. Which aspect, each statistical test is testing should be clearly stated. Authors have used Chi-sqaure test throught the analysis, and this is the reason why they did not subject their data to normality tests. If they want to use tests like t-tests or AnoVA then they would be required to convert their data into proportions. Proportions then have to be logit-transformation. Thus, I personally ask them to continue with Chi-Squre test.

However, the methodology should explain why they used "instantaneous scan sampling" instead of other recording/sampling rules. Why not focal sampling, because some groups had only two individuals. How did they compute diet composition. How and where in the manuscript did they analyze feeding preferences and food availability. How did they collect data related to food availability. How food plant parts were defined. Note-if they report on feeding preferences, then they should show how they compute food selection ratios and how to interpret them.

Validity of the findings

Some are valid-For example, they have clearly shown some temporal dietary flexibility.

But in my opinion, authors need to 1) compute the overall diet composition (plant parts) as they have done for the seasonal diet composition. 2) Overall and seasonal plant species consumption by guereva. 3) Food plant and plant species selection/preference- this will require to report food plant species abundance/availability in the four groups' home ranges.

For eg. We still do not know which plant species are preferred based on their availability. A key finding which is important for future monitoring.

Additional comments

General comments
The manuscript requires a very thorough revision before it can submitted to PeerJ. I recommend a comprehensive revision of the manuscript. The English in the entire manuscript should be improved to ensure that an international audience can clearly understand your text (from the abstract to the conclusion). I suggest a native English speaker can do this as I am not an English Speaker. Coherence should be maintained throughout the manuscript from the Abstract to Conclusion.
My Major Concerns: The authors should re-analyze the data for the eight plant species eaten (i.e. feeding records per plant species eaten for the 12-month duration and/or seasonally).
The authors in the title have mentioned “feeding preference” but it is not clear how this was addressed throughout the manuscript. To do this they need to compute plant part selection ratios or plant species selection ratios but I have not seen this anywhere in the manuscript. Again to do this, they need to determine food availability of plant species or phenological patterns of plant parts like young leaves, flowers, mature leaves etc-as this will enable them to compute plant selection ratios. However, I did not think if they recorded data to determine food availability in the groups’ home ranges. If they have not done this, then they may consider changing the title.
Specific Comments
Comment1: Please include the correct name of the study species. Is this a black-and-white colobus or guereza or colobus monkey? I recommend authors either to replace “colobus monkey” in the title with “Guereza” or Black-and-white colobus”. Or they contact an expert on this species or use the common name from the IUCN website.
Comment 2: The TITTLE includes an aspect of “feeding preferences” but surprisingly I have not seen anywhere throughout the manuscript selection ratios/preferential ratios have been computed. I may conclude this part has not been comprehensively done. On the other hand, the interesting part is the observation of seasonal differences found with regard to young and mature leaf consumptions (mature leaves eaten more in dry than wet while young leaves eaten more in wet season) suggesting some flexibility. Thus I may suggest this very simple title “Dietary composition and flexibility of Guereza (Colobus guereza) in Maze National Park, Ethiopia”. This title will also include food selectivity if the authors wish to continue with the analysis (i.e. analysis of food selectivity/preference).

COMMENT 3: ABSTRACT. In the abstract avoid words like “plant types” should be “plant species”.

THE FOLLOWING IS MY EDITS IN YOUR ABSTRACT- The following is may tentative abstract for the authors

Knowledge of feeding ecology is essential for effective management of a primate and its habitat. The black-and-white colobus (Colobus guereza) is a predominantly folivorous monkey occurring in different parts of Africa, including the Maze National Park in Ethiopia. Despite many studies on its feeding ecology and dietary flexibility, there is no study that was carried in the in the Maze National Park (MZNP) up to date. The aim of this study is to determine its dietary composition and flexibility of the study species in the park. Instantaneous scan sampling method was used to collect feeding data during October 2021-September 2022. We followed monkeys from 6:30 to 10:30, collecting activity data between 5 minute intervals during 10-minute scan duration. If the activity of monkey was feeding, food plant part and species consumed were also recorded. Overall, eight plant species and non-identified invertebrates have been recorded to be consumed by guerezas in the park. We also found that young leaves were consumed more in the wet while mature eaten more the dry season, suggesting that the guerezas in the park exhibit temporal dietary flexibility as reported by other researchers elsewhere. The observed dietary flexibility may be partly due to seasonal changes in availability of food plant parts in the studies groups’ home ranges in the park. Our results suggest that maintaining the riverine habitat along the Maze River, where most of essential plant types and feeding activities were recorded was crucial for the conservation of Colobus.

LINE 35-36: Revise the key words according to the Peer J instructions.

LINE 38-86. INTRODUCTION: Before line 38, you can begin your introduction by briefly explaining why studies of feeding ecology are important for primate conservation. Then continue with line 38. The introduction should include a research gap –impliying why you have decided to embark on studying feeding ecology of the guerezas especially in the Maze Park. After this then follow up with your study aims at line 77. Generally, the whole introduction will need some sort of polishing the language and making the sentences or pargraphs coherent.

Line 128-134: PLEASE REPHRASE THIS SECTION. The original reference is “Altmann, J. (1974). Observational study of behavior: sampling methods. Behaviour, 49(3-4), 227-266”

I have suggested this paragraph for Line 128-134.
We collected feeding data for During the 12-month period between (October 2021 and-September 2022, covering the dry season from October 2021 to February 2022 and wet season from April to August 2022. We collected these data we collected feeding activity data from individuals using instantaneous prompt scan sampling (Altmann, 1974, 2009) between 5-minute intervals and 10-minute scan durations from 6:30 am to 10:30.. For the dry season, we collected data from October 2021 to February 2022, and for the wet season, from April 2022 to August 2022. The feeding data were collected through direct observation from proper viewpoints (Altmann, 2009). This was watching of the Colobus guereza for a fixed period of 10 minutes with 5 minutes interval from 6:30 to 10:30 within naked eye and binocular depending on the distance between the observer and targeted group of the study species (Fashing et al., 2007).

LINE 128-135: How many days did you follow each group for feeding data collection?. Briefly explain why you did not follow groups from dawn to dusk but only in the morning sessions like 6:30-10:30 as shown in your methodology. Sometimes, primates exhibit diurnal dietary flexibility/variations so missing the afternoon or early evening data may make your conclusion not a decisive or representative conclusion, especially with regard to plant species eaten. Also briefly explain whether these monkeys were habituated.

LINE 135 “Target groups” and its associating paragraphs should come before Line 126 Data collection.
Use this sequence please below:

MATERIALS AND METHODS
-Study area
-study study groups
-Data collection
-Data Analysis
- Ethical Note).

NOTE: Why the study groups are very small in size?? Authors have to provide some explanations for this!!!!

LINE 142-147: This should be moved at the ending sentence at line 134, then you can do some rephrasing/language polishing. Change “plant type” into “plant species” in line 142 and parts of plants into “plant parts” and “the diet components other than the plant materials were noted” into “other animals consumed were recordes” . Line 142, change “In each exploration interval” into “During scans” or “At each scan sample”……….

Line 143 when you say “growth patterns” what do you mean” do you mean “growth forms” like trees, shrubs, climbers etc. If you mean so, then change “growth patterns” into “growth forms”.

Line 144-145. Please you need to define food plant parts young leaf, matured leaf, shoot, flower, fruit, bark and unidentified invertebrates. For example what do you mean young leaves? What do you mean mature leaves? What do you mean, shoot etc. You can define them within the text or make a table and define them.

Line 150-157: Change the word “feed” to “food”. This should be revised through out the manuscript.

Line 159-163 You can remove this whole paragraph
Line 164 You can change this sentence into “Annual and seasonal diet composition of guerezas”. Then compute the overall diet composition plant parts eaten for all 12-month periods. Then follow up this with the paragraph seasonal diet composition as you have reported in this section.

Then make another subtopic titled “Food plant species consumed. Here you may compute the overall feeding records (i.e. for 12-month period) and seasonal feeding records of each eaten plant species (irrespective of plant specific parts). IF you did a vegetation sampling to determine food availability, then you can compute the %ge stem density or basal area or crown covers to estimate food selection ratios in order for you compute feeding preferences (which you have not reported anywhere throughout the manuscript).
Line 192-193 –Move the first sentence of this section into the Methodology section (see Mekonnen et al., 2017; 2018 on feeding flexibility of Bale monkey”. Or also Kibaja et al., 2023 red colobus monkey.

The entire discussion should be revised according to the changes made in the result section.
The conclusion also should be revised according to the results and include a way forward.

·

Basic reporting

This paper provides useful information on the feeding behaviour of Colobus guereza in a specific park with marginal habitat for the species. The data is probably collected in a good quantitative way and there is probably enough data available, but the presentation of the data is not sufficiently informative for publication.
The raw data files presented are not full raw datafiles but instead contain summary tables. We would like to see your daily records, which animal was sampled in which record and how many records there are per month and per day and per group and individual etc. I have given a lot of feedback on the PDF of the manuscript and I hope it really helps you prepare the paper in a way that makes it acceptable for publication, but I think it would be worth talking to somebody with extensive experience in publishing in international journals.
Kind regards

Experimental design

The design seems okay, although we would normally have a longer time period in between scan samples. The trouble is that data analyses should have taken account of interdependencies within the dataset. For example, samples from the same group should be considered related to each other.

Validity of the findings

We need more information, the basic findings are valid, but the way it is analysed and presented does not allow us to truly see more interesting patterns.

Additional comments

See above

---

## Round 0.2 · Major Revisions

I'm grateful to the Reviewers for their exceptionally detailed comments to improve the manuscript and their level of support and. in effect, mentorship. Both have gone out of their way to provide supportive feedback (more than usually would be expected) and have shared specific comments and helpful edits on the manuscript document. Please pay attention to the guidance regarding the format of the manuscript, in particular, the reviewers comments to improve the writing structure (and also links to writing guidance).

I agree with both reviewers that a clearer explanation needs to be provided about how the statistical analyses were conducted and for the analyses to be properly completed.

·

Basic reporting

The manuscript should be improved, especially in the Discussion section. Authors can also improve some results and introduction. The reviewer has made some edits throughout these sections. The last version should contain all references cited in the texted, be listed in the reference section.
My edits have made the manuscript more focused and concise.

Experimental design

Include how selection ratios were computed. And how these selection ratios can be interepreted (but very briefly)

Validity of the findings

Valid bse data were collected for 12 months. Despite the length of data collection, authors have recorded only a few plant species eaten, so this is worth for conservation.

Additional comments

Lines 19-20 I recommend this sentence “ We used instantaneous scan sampling method to collect feeding data from September 2021-August 2022”. I have already edited it in the abstract (see the attached manuscript).
Lines 20-21 I would delete this sentence “We predicted that the Colobus guerezaís preference of food type 21 is determined by the season and the accessibility of food sources”. Of course I have deleted it in the attached manuscript.
Lines 23-24 I would also delete this sentence “We computed monthly and seasonal difference 24 of diet components consumed by the species using chi-square test”. See attached reviewed manuscript.
Line 26 delete the “during” as I did in the attached/reviewed manuscript
Line 158-165, I have deleted the entire text. See attached manuscript. Be very straight to the point.

INTRODUCTION. Ihave edited it by introducing new sentences. Some texts were cut and pasted from here and there in the section. I have made it very focused. The authors can polish it better, but I have given them some plan.
METHODOLOGY. This section is generally good. But the authors have to explain very briefly how they computed selection ratios here. And how these selection ratios can be interepreted (they can consult Mekonnen et al., 2018 dietary flexibility of Bale monkeys).

RESULTS. The authors have to be very focused here, they should focus or stick to their objectives. I have re-written this section, so it is up to the authors to polish or improve my edits/write-up. Otherwise they can introduce the monthly data for food plant species in a tabular form as they did for Table 5 (in my attached manuscript-reviewed).

DISCUSSION: It was very long but very poorly linked to the study objectives. I have tried to re-write this section (using my opinions and the authors text). They can improve whate I have written there.

CONCLUSION. I have also introduced some text in the beginning.

TABLES AND FIGURES: I have recommended some new TABLES at the end of the manuscript. Authors can see them if they can make some improvements.
In my opinion the manuscript will be very focused and very concise.

·

Basic reporting

There are clear weaknesses in this aspect. See my full report.

Experimental design

This is fine.

Validity of the findings

The statistical analyses are not well explained and it is hard to be sure they were done correctly.

Additional comments

The revision is improved in many ways, but there is still a lot to be done on this paper before it can be published.
Most importantly, I am not provided enough evidence to be sure that the statistical analyses were done correctly and don’t understand why some analyses were done. Secondly, the food preferences analyses have not been fully done yet. Finally, the discussion is still far too long and repetitive. Effectively, the whole discussion is about how diets vary over the year according to what is available. This does not need to be said more than once, so please reduce this.
The writing is still not at the right level either. I understand this can be hard but there is now great software available, like ChatGTP that will help you rewrite your text into better written English. Please do check that the outcome does not change the meaning of your text of course.

Specific Comments:
it is always a strong opening when you are able to put into the first paragraph: 1-2 sentences context/ the big issue, 1 sentence of 'what needs to be done to address the issue, 1-2 sentences of what we already know (you are missing this), and 1-2 sentences of what is not known and needs addressing. Try to have more information non the focus of this study in that final sentence.
e.g.:

Habitat change and climate change together are greatly reducing the access species have to their most important food sources. Effective species’ conservation management strategies, such as habitat management and supplemental feeding programs, therefore, depend on a good understanding of species’ nutritional needs and dietary preferences (Ramesh & Downs, 2013). Feeding ecology studies can be used to identify crucial food resources and their seasonal or annual availability (Sengupta et al., 2015) and to quantify the effects of habitat change, loss and fragmentation on animal populations (Irwin et al., 2010). Furthermore, these investigations can indicate how animals adjust their nutritional preferences in response to environmental changes (Gogarten et al., 2012). Primates are among the most endangered mammals and their general feeding ecology is well understood, but feeding ecology is highly site- and species-specific. To guide ecological restoration efforts and inform sustainable forest management practices, we need site-specific information to ensure the availability of critical food resources for primates (Ganzhorn et al., 2017).

You don’t have to copy that exactly, but I hope this explains what I mean by structuring the first paragraph of your paper. This article explains it a little The 5 pivotal paragraphs in a paper | Dynamic Ecology (wordpress.com) or this one Intro-Discussion-and-Lit-Review-Writing-Resource.pdf (usu.edu). You don’t give the aim of the study in the first paragraph in our field, instead just narrow down to the focus of the study but leave the aim until the last paragraph of the introduction.
L63: if it is least concern then there does not seem to be a reason to concern ourselves. So the ‘Therefore’ does not read well. Please explain more clearly why it is still urgent to conserve their habitat to safe the species. Maybe because it is highly fragmented already and the small groups and small populations are at risk of disappearance due to Allee effects.
L73: if this preference has already been shown, please give the citation for that paper.
L89-96: I do not like seeing the results already at the end of the introduction, but some journals and authors prefer it so I leave this up to the editor to comment on. I would personally remove these.
L170: If you observe them for 10 minutes, what is your reference for activity? It is not instantaneous? Normally we have shorter ‘instantaneous’ observations to determine activity: you see an individual, wait 5 seconds and record their activity at that moment in time. Then you move to the next individual. So in most cases if there are just 3 individuals, that is probably completed in less than 10 minutes. For that sample to be independent from the preceding sample, we should wait some time. Normally, therefore, the observation period (in your case 10 minutes max.) would be shorter than the period ‘in-between samples’ (in your case the 5 minutes). Otherwise, they are just doing the same thing they were doing the first time around so you have the issue of interdependence among sampling points. In the end, I doubt that this statistical issue actually affected the data much so I won’t make a point of it but it is worth looking into your sampling method a bit more. I suspect that effectively, your ’10 minutes’ was much shorter in most cases. Unless you looked at each individual for a whole of 10 minutes and then wrote down activity as the thing they were doing the most in those 10 minutes. Happy to discuss this with you after the paper review is completed for future studies.
L217: ‘sections’ is not the normal word to use, you want to say plant parts.
L220: you say that the consumption related to availability. If you make that statement, you must also here show your statistical analysis of preference. So give us the information on tree species relative abundance and plant part abundance over the months/ seasons.
Figure 2: this is not really easy to read, I suggest you replace it with a table.
L220: you refer to Figure 2 but actually the analyses about wet versus dry season are in tables 1 and 2, not in Figure 2 where we don’t know which month is wet or dry.
L225: “We found in the dry season, mature leaf accounted for the majority (43.61%) of colobus. Wet” are you referring to the dry or wet season?
L229 “We found that the seasonal consumption of food items varied consumed between similar food categories revealed significantly between months (Young leaf: Chi2 = 405. 140, df = 1, p < 0.05; Mature leaf: Ç2 = 651. 563, df = 1, p < 0.05; Fruit: Ç2 = 105. 593, df = 1, p < 0.05; Shoot: Ç2 = 125. 063, df = 1, p < 0.05).” I am a little confused as to what you actually tested here. In this analysis here, did you compare use across the months (“between months”)? Those Chi2 values are very high.
Tables 1 and 2 give us percentages. I assume you correctly did the Chi2 test on actual counts rather than percentage values and corrected the expected values of the Chi2 test according to the number of samples collected in that month, since one cannot do a Chi2 test on percentages. For that reason, it may be more informative to present a table that shows observed and expected counts of feeding records for each item per season or per month.
“We also determined showed increasing variation between food components utilized during within each of the study seasons (Dry: Ç2 = 1629.320, df = 7, p < 0.05; Wet: Ç2 = 3620.960, df = 6, p < 0.05).” Again the Chi2 values are extremely high, and you have a df of 7 in the dry and 6 in the wet season. So I would like to see more evidence on how the Chi2 test was executed. A table or figure showing observed versus expected would help us understand the actual analyses you did. Also, if you compared all the categories, there would be a lot of cells with values less than 5 in the Chi2 test, what did you do to avoid that? Typically we merge categories with low values.
Tables 1 and 2: please merge these to one table (or a and b). Please also include in the table caption, which months constitute dry versus wet season.
L234: You don’t need to repeat why you did it.
L237: You state that the variation was analysed, but then proceed without showing the results of those analyses. I feel that what you say here is actually what was analysed in the preceding section, so it should not be mentioned in this section.
L239-240: either gives us information about months or seasons, this sentence mixes it all up and is confusing.
L239: I think you should refer to Table 3 here regarding your analyses of items per month?
L241: “The number of observations for feeding among study months showed significantly varied (Ç2 = 43.920, df = 11, p < 0.05) (Table 3).” So the df is 11 because you compared 12 months here I assume. What I don’t understand is how ‘the number of feeding observations’ is important to analyse. Are you saying that during some months they spent less time feeding than other months? Then tell us what they were doing instead of feeding. You have that information. Myabe in some months there was more resting whilst in others there was more moving/ travelling?
The Chi2 analyses presented in L241 does not explain the analyses presented in Table 3. What were you comparing in Table 3? There is an analyses in each individual month but I don’t understand what is analysed, especially not since the DF value changes between months. What is represented by the rows/ columns in your analyses that df could be either 7 or 2 depending on the month? Again your Chi2 values are extremely high, which sometimes indicates that the wrong things have ended up in the actual analysis.

L243 section: The feeding preferences (plant item and plant species) are not clearly analysed statistically. You can easily again do a Chi-2 in which you use the relative abundance of each tree species to calculated the expected count of feeding records from that tree species and compare it to the observed count of records from that species. I would only do this for the top species so that no cells are <5 counts in the Chi2 analysis.
Until that analyses is done and we see a graph or table showing observed versus expected, it is not really analysed.
L271: start the discussion with a clear statement of the main context of the study.
Thus, population biology and ecology
L275: “To make up the majority of herbivore.s diet, it is necessary to assess the quantity and quality of the most and least desired plant species (Ego et al., 2003). The largest selection ratio for plant parts shows a desire for the food items offered by the plant species, whereas a low selection ratio implies a dislike (Mekonen & Hailemariam, 2016; Fashing et al, 2007).” These sentences do not fit into the first paragraph of the discussion. I suggest they go into the introduction or somewhere. To be honest, I do not think anybody needs explaining what ‘preference’ means, it just needs to be calculated in your study and you did not truly analyse this yet.
L281: the tables show what they are eating but do not show what they are eating in relation to how common those items are. So stating that they ate more YL when there was more YL and referring to Tables 1 and 2 does not work. You could improve Tables 1 and 2 and actually give us for those months the representative value of each food part: so the percentage of your sampling trees were bearing those plant parts.
L296: ‘these’ what do you mean? These papers? This study’s results? I don’t know what that sentence is trying to say.
L301 “and during the dry season and vice versa”
The discussion is overly long still and there is a lot of repetition, almost every section states the same thing: there is dietary variation across seasons and this depends on availability of food. We don’t need to be told that more than once, so please reduce your discussion considerably. In L346 there is another section that basically again talks about dietary variation. Please reduce.



Specific comments
Please add/remove/change as follows or similar to:
L15: Eastern Africa
L16: “no up-to-date data that was carried out on feeding”
L17: “To better understand this, We randomly”
L20: “We predicted that the Colobus guereza.s preference of food type is determined by the season and the accessibility of food sources.” Move this to L18, right after stating the aim.
L28: “We also found that young leaves were consumed more (75.31%) in the wet while mature eaten more (43.61%) in the dry season” More than what? Please don’t just give the percentage value for one thing, tell us also what the mature leaf consumption is in the wet season and the young leaves in the dry season. Or is this % of all their consumptions of young/mature leaves? The % information needs to be clarified.
L30: “The diet components consumed by the study species shown significantly varied between seasons.”
L99: “River, which is a major habitat of the target species.”
L116: “composition, that the portions of this text were previously published as preprint”: you don’t need to worry that some of the site description is similar to what was already published. This section is often ‘self-plagiarised’ because there are just so many ways in which you can describe a site. Please replace with: “composition, for more details on the site see….”.
L198: ‘Nevertheless’ replace with “Although”
L217: “and utilized by one or more of its section during in the dry season, while six” this sentence is confusing. What are sections here? And how can a section utilise anything?
L221: plural of ‘leaf’ is ‘leaves’ (I agree it is confusing

---

## Round 0.3 · accepted · Accept

You have addressed all of the reviewers' comments and I'm very happy to accept the manuscript. There are a few minor typos in the text and Reviewer 2 has kindly annotated these in the attached PDF. Once these have been amended the manuscript will be ready for publication. I would like to thank you for your patience and for engaging with the review process.

·

Basic reporting

This new revised manuscript reads really well. I made just a (very) few comments regarding typos.

Experimental design

Good.

Validity of the findings

Good.

Additional comments

The authors have done a great job revising the manuscript, it is now a very nice piece of work.